# Community use of oral antibiotics transiently reprofiles the intestinal microbiome in young Bangladeshi children

Andrew Baldi[1,2] ✉, Sabine Braat [1,3,4], Mohammed Imrul Hasan[1,2,5], Cavan Bennett [1,2], Marilou Barrios [6], Naomi Jones[1], Gemma Moir-Meyer[1,2], Imadh Abdul Azeez [1,2], Stephen Wilcox[6], Mohammad Saiful Alam Bhuiyan[5], Ricardo Ataide[1,4], Danielle Clucas[1,2,7], Leonard C. Harrison [1,2], Shams El Arifeen[5], Rory Bowden [2,6], Beverley-Ann Biggs[4,8], Aaron Jex[1,2,9] & Sant-Rayn Pasricha [1,2,7,10] ✉

Antibiotics may alter the gut microbiome, and this is one of the mechanisms by which antimicrobial resistance may be promoted. Suboptimal antimicrobial stewardship in Asia has been linked to antimicrobial resistance. We aim to examine the relationship between oral antibiotic use and composition and antimicrobial resistance in the gut microbiome in 1093 Bangladeshi infants. We leverage a trial of 8-month-old infants in rural Bangladesh: 61% of children were cumulatively exposed to antibiotics (most commonly cephalosporins and macrolides) over the 12-month study period, including 47% in the first 3 months of the study, usually for fever or respiratory infection. 16S rRNA amplicon sequencing in 11-month-old infants reveals that alpha diversity of the intestinal microbiome is reduced in children who received antibiotics within the previous 7 days; these samples also exhibit enrichment for *Enterococcus* and *Escherichia/Shigella* genera. No effect is seen in children who received antibiotics earlier. Using shotgun metagenomics, overall abundance of antimicrobial resistance genes declines over time. Enrichment for an *Enterococcus*-related antimicrobial resistance gene is observed in children receiving antibiotics within the previous 7 days, but not earlier. Presence of antimicrobial resistance genes is correlated to microbiome composition. In Bangladeshi children, community use of antibiotics transiently reprofiles the gut microbiome.

Antimicrobial resistance is a leading global health threat, rendering vital antibiotic medicines ineffective against life-threatening infections[1]. A major proposed driver of antibiotic resistance in low- and middle-income countries is poor antimicrobial stewardship fostering high rates of inappropriate use of antibiotics[1,2]. Antibiotics are among the most prescribed medications. Over 73 billion doses of antibiotics were used across 71 countries in 2010[3,4]. Use is growing fastest in low- and middle-income countries, where on average 52% of

patients attending health care are prescribed antibiotics[5]. Inappropriate or subtherapeutic use of antibiotics may drive antimicrobial resistance via selection for bacteria with natural resistance to antibiotics, including mutant clones containing resistance genes, or which have acquired resistance via horizontal gene transfer[6].

Asia is a region where limited antimicrobial stewardship may especially contribute to an environment facilitating the emergence of antimicrobial resistance[7]. For example, antibiotic medications may be

available 'over the counter' without prescription, and patients may purchase medications for themselves or their children without first establishing a diagnosis or even visiting a health practitioner or facility[8]. In Bangladesh, 'drug shops', which may be unlicensed and staffed by unqualified personnel who provide medical advice, are commonly accessed to procure antibiotics[9]. Exposure of children to antibiotics is high. Multinational cohort studies across South Asian, Latin American and sub-Saharan African countries have measured community use of antibiotics by young children under two years of age. Usage is highest in children living in South Asia, where children in Bangladesh received 10.9 courses of antibiotics per year and almost all children received antibiotics within the first six months of life[10].

Systematic reviews indicate that antibiotics used in primary care for respiratory or urinary infections may foster antimicrobial resistance[11]. In Bangladesh, studies evaluating clinical isolates have identified pathogenic bacteria such as *Salmonella typhi*, *Escherichia coli,* and *Staphylococcus aureus* resistant to commonly used, cost-effective antibiotics, with rates of resistance among the highest in the world[12,13]. For example, a systematic analysis of the global burden of antimicrobial resistance indicated that in Bangladesh, *E. coli* resistant to third-generation cephalosporins and/or fluoroquinolones represented 60-70% and ≥80% of isolates respectively, while the prevalence of methicillin resistance in *S.* aureus is 40-50%[14]. Most studies evaluating the effects of antibiotics on microbial composition and antimicrobial resistance have used clinical isolates, were set in inpatient or high-acuity clinical settings, and used bacterial resistance functional testing in vitro. More broadly, complex interactions between the environment (including water, sanitation and hygiene conditions, and livestock and wildlife) may influence antimicrobial resistance[15,16].

Therefore, a critical question is whether and how antibiotic use in the community in low-income settings, where exposure to infections and other environmental drivers is intense, directly influences microbial composition and antimicrobial resistance, particularly in children. We thus sought to determine whether, in a rural South Asian setting, antibiotic exposure through the community reprofiles the gut microbiome and influences the carriage of antimicrobial resistance genes in children.

To achieve this aim, we leverage a prospective cohort study of children aged 8 months nested within a large field randomized controlled trial that aimed to evaluate whether iron interventions influence child development set in rural Bangladesh to measure patterns of antimicrobial use over a 12-month period[17]. We apply unbiased genomic techniques to evaluate the effects of community use of oral antibiotics on the composition and antimicrobial resistance profiles of the gut microbiome.

## Results

We recruited 8-month-old infants to the BRISC (Benefits and Risks of Iron InterventionS in Children) trial in Rupganj Upazila (a rural area about 50 km from Dhaka) in Bangladesh[17] and followed for 12 months. Of these 3300 children recruited to the main trial, the final 1093 recruited to the trial (between September 2018 and February 2019) were invited to participate in this microbiome sub study. The baseline (at 8 months of age) characteristics of these children (923 of whom provided at least one stool sample) are summarized in Table 1. In this cohort, at baseline, the median age additional foods were added to breastfeeding was 6 months; 19.4% of infants were from families with food insecurity, and 23.6% of children exhibited stunting and 7.4% were underweight.

Participants were visited weekly during the 13-week intervention period (iron, multiple micronutrient powders – MNPs – and placebo) and then monthly for 9 months (Fig. 1A). Stool was collected at baseline, after 13 weeks intervention (midline), and again after a further 9 months (endline). 923 and 319 samples from baseline, 796 and 319 samples from midline, and 578 and 315 samples from endline were processed for 16S rRNA amplicon sequencing and shotgun metagenomics, respectively (Fig. 1B).

**Table 1 | Participant baseline characteristics**

| | Total n = 1093 (%) |
|---|---|
| *Treatment arm* | |
| Iron | 364/1093 (33.3) |
| MNPs | 368/1093 (33.7) |
| Placebo | 361/1093 (33.0) |
| Compliance 70% or higher | 820/1093 (75.0) |
| *Sex* | |
| Female | 539/1093 (49.3) |
| Male | 554/1093 (50.7) |
| *Union* | |
| Bhulta | 349/1093 (31.9) |
| Golakandail | 375/1093 (34.3) |
| Rupganj | 369/1093 (33.8) |
| *Maternal education* | |
| No education | 38/1093 (3.5) |
| 1–8 years | 563/1093 (51.5) |
| 9–12 years | 452/1093 (41.4) |
| >12 years | 40/1093 (3.7) |
| *Paternal education* | |
| No education | 83/1093 (7.6) |
| 1–8 years | 578/1093 (52.9) |
| 9–12 years | 374/1093 (34.2) |
| >12 years | 58/1093 (5.3) |
| *Maternal occupation* | |
| Unemployed | 1061/1093 (97.1) |
| Unskilled job | 10/1093 (0.9) |
| Skilled job | 22/1093 (2.0) |
| *Paternal occupation* | |
| Unemployed | 11/1092 (1.0) |
| Unskilled job | 158/1092 (14.5) |
| Skilled job | 874/1092 (80.0) |
| Other | 49/1092 (4.5) |
| *Wealth index quintile* | |
| Quintile 1 (relative poorest) | 203/1093 (18.6) |
| Quintile 2 | 199/1093 (18.2) |
| Quintile 3 (relative middle) | 237/1093 (21.7) |
| Quintile 4 | 221/1093 (20.2) |
| Quintile 5 (relative wealthiest) | 233/1093 (21.3) |
| Household Food Secure status[a] | 868/1085 (80.0) |
| Age extra food in addition to breastfed (months) | 6.0 (5.0–6.0) |
| *Child Growth* | |
| Stunted[b] | 258/1091 (23.6) |
| Underweight[b] | 85/1091 (7.8) |
| Wasting[b] | 10/1091 (0.9) |

Data are presented as mean (SD) or median (IQR) for continuous measures, and n/total (%) for categorical measures.
*MNPs* multiple micronutrient powders.
Percentages may not total 100 because of rounding.
[a]Food secure was defined 'no' or 'rarely' to question 1 and 'no' to questions 2–9 on the Household Food Insecurity Access Scale.
[b]Z-scores were calculating using the 2006 World Health Organization Child Growth Standards.[50] Stunting was defined as a length-for-age z-score <−2. Underweight was defined as a weight-for-age z-score <−2. Wasting was defined as a weight-for-height z-score <−2.

Antibiotic use was assessed at each home visit: a field worker directly enquired about antibiotic consumption, and asked to see the container to confirm the medication for coding. Across the sub study cohort, over the 3-month intervention period, 47% of children received

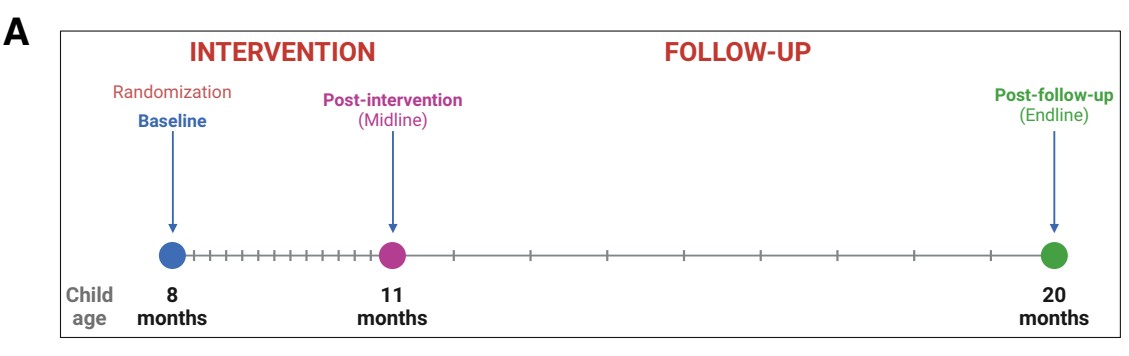

at least one antibiotic; over the entire 12 months this figure was 61% (Fig. 1C). These antibiotics mostly comprised cephalosporins, macro-lides, nitroimidazoles (all metronidazole), penicillins (e.g. amoxicillin), quinolones (all ciprofloxacin), and co-trimoxazole (Fig. 1D). The most reported symptoms associated with the children's antibiotic use were fever and respiratory symptoms (Fig. 1E).

We first used 16S rRNA amplicon sequencing to measure the effects of antibiotic usage on stool microbiome alpha diversity (indi-cating the distribution and heterogeneity of taxonomic abundances, measured by the Shannon and inverse Simpson indices). Firstly, we established that as expected, alpha diversity increased between 8 and 20 months (Supplementary Fig. 1)[18]. Next, we explored the effects of

**Fig. 1 | Study design and antibiotic use. A** BRISC trial schema showing assessment and sampling time points. Vertical lines represent scheduled study visits. **B** Flow diagram outlining stool samples for 16S rRNA and shotgun metagenomic sequencing. **C** Cumulative incidence of parent-reported antibiotic use among sub study participants as recorded weekly (Weeks 1–13) and monthly (Months 1–9) thereafter. **D** Reported antibiotic use during the intervention period by class (where metronidazole, ciprofloxacin, and co-trimoxazole were recorded as standalone medication as they were the only medication recorded in their respective class). Sum of

percentages exceeds 100 due to instances of multiple antibiotics being documented at single visits. **E** Prevalence of fever and infective symptoms reported during the intervention period (where multiple symptoms could be recorded in a single weekly visit). **A**, **B** created with BioRender.com released under a Creative Commons Attribution-NonCommercial-NoDerivs 4.0 International license (https://creativecommons.org/licenses/by-nc-nd/4.0/deed.en). Source data for **C**–**E** are provided in the Source Data file.

**Fig. 2 | Alpha diversity and microbiome differential abundance using 16S rRNA amplicon sequencing data. A**–**C** Violin plots illustrating taxonomic alpha diversity in relation to antibiotic use in Weeks 1–4 (81 with antibiotic use from a total of 671) (**A**), Weeks 9-12 (158/771) (**B**) and Week 13 (44/727) (**C**) (16S rRNA data). Each dot represents an individual sample, grouped by defined antibiotic use category (yes/no). Group differences were calculated using pair-wise ANOVA for each diversity measure. Center lines denote median value, with rectangles showing 25–75th percentiles. The violin outlines the distribution of the data, with wider sections

representing a higher probability that samples in the dataset will have the corresponding value and narrower sections representing a lower probability. **D**–**F** Volcano plots illustrating differential abundance of genera in relation to antibiotic use in Weeks 1–4 (81/671) (**D**), Weeks 9–12 (158/771) (**E**), and Week 13 (44/727) (**F**) (16S rRNA data). Each figure shows log₂-fold change on the x-axis and the -log₁₀(FDR-adjusted *P*-value) on the y-axis. The horizontal red line indicates an adjusted *P*-value 0.05 (calculated using the Benjamini–Hochberg method). Source data are provided in the Source Data file.

antibiotics on stool alpha diversity at the midline timepoint (~11 months of age), the timepoint for which we had highest resolution of information on preceding antibiotic consumption (i.e., last 7-days for every week). Overall, use of antibiotics within the previous 13 weeks was not associated with changes in alpha diversity based on there being no differences in Simpson and Shannon indices compared to those without any use of antibiotics (Supplementary Fig. 2A). We next disaggregated the analyses by timepoint of exposure to antibiotics. Alpha diversity was

lower in samples from infants who had received an antibiotic in the preceding week compared to those who had not ($P < 0.001$ for Shannon index and 0.011 for inverse Simpson index). However, there were no significant differences in alpha diversity in samples from infants receiving antibiotics in Weeks 9-12 or in Weeks 1–4 of the intervention period compared with children who had not received any antibiotics in Weeks 9–12 (for Weeks 9–12 analysis) or who had no antibiotic use for the whole intervention period (for Weeks 1–4 analysis) (Fig. 2A–C).

We next sought to establish the effects of antibiotics on intestinal microbiome composition.

Differential abundance analyses of the 16S rRNA amplicon analysis revealed enrichment for *Enterococcus* and *Escherichia/Shigella* genera in samples from children receiving antibiotics within 7 days of collection (log$_2$FC 2.076, SD 0.439, FDR-adjusted $P < 0.001$ for *Enterococcus and* log$_2$FC 1.154, SD 0.330, FDR-adjusted $P = 0.012$ for *Escherichia/Shigella*), but this was not seen with earlier antibiotic use (Fig. 2D–F). We used shotgun metagenomic analysis in a subset of sub study participants to achieve a higher resolution of classification and to identify antimicrobial resistance genes. We determined differential abundance of species, and mapped reads to the Comprehensive Antibiotic Resistance Database (CARD) reference markers to identify abundance of antimicrobial resistance (AMR) genes (ARGs)[19]. We first evaluated overall AMR gene abundance in samples from all three time points (8, 11, and 20 months). There was a significant decline in overall AMR gene abundance between 8 and 20 months of age (mean gene abundance at 8 months 4356 Reads Per Kilobase per Million mapped reads [RPKM], standard deviation [SD] 3215 compared to 3208, SD 2596 at 20 months, $P < 0.0001$) (Fig. 3).

Using shotgun metagenomics, we observed an increased relative abundance of *Enterococcus faecium* in children who used antibiotics in the preceding week (log$_2$FC 0.761, SD 0.156, FDR-adjusted $P < 0.001$). This group showed higher abundance of ARO:3002556 compared to those who had not used an antibiotic in that week (log$_2$FC 0.995, SD 0.269, FDR-adjusted $P = 0.040$). ARO:3002556 corresponds to the *Enterococcus faecium* chromosomally encoded aac(6′)-Ii aminoglycoside acetyltransferase gene, which confers aminoglycoside resistance (card.mcmaster.ca/ontology/38956/)[20]. Antibiotic use overall (Supplementary Fig. 2B-C), or at earlier time-points (in Weeks 9–12 or Weeks 1–4 of the intervention period) was

not associated with any differentially abundant species or AMR genes compared to no antibiotic use in Weeks 9-12 (for Weeks 9-12 analysis) or no antibiotic use for the whole intervention period (for Weeks 1–4 analysis) (Fig. 4A-F). To explore whether the effect of antibiotic use on AMR gene abundance was influenced by study intervention arm (i.e., iron/MNPs or placebo), we conducted an additional analysis of interaction. This indicated no evidence of an interaction between iron interventions and effect of antibiotics on AMR (Supplementary Fig. 3A). There was also no evidence of an interaction between iron interventions and the effect of antibiotics on microbiome composition (Supplementary Fig. 3B).

Finally, we sought to evaluate whether the abundance of AMR was driven by microbiome composition. Using shotgun metagenomic data, we examined the association between abundance of particular genera of interest and overall AMR gene abundance at both the midline and endline timepoints (i.e. at age ~11 months and ~20 months). At the 11-month timepoint, *Escherichia* showed a highly positive correlation (Spearman $r = 0.86$, $P < 0.001$) with overall AMR gene abundance. In contrast, *Enterococcus* exhibited a weaker correlation (Spearman $r = 0.16$, $P = 0.009$), although this association appeared to strengthen in children who had received an antibiotic in the last 7 days: Spearman $r = 0.44$, $P = 0.066$) (Fig. 5A). At the 20-month timepoint, using endline samples, total AMR gene abundance and *Escherichia* relative abundance were moderately correlated (Spearman $r = 0.67$, $P < 0.001$), while AMR correlation with *Enterococcus* was also observed (Spearman $r = 0.23$, $P < 0.001$) (Fig. 5B). A sensitivity analysis following removal of two midline samples and one endline sample with *Escherichia* relative abundance >90% did not change our results. We also observed a reduction in overall abundance of *Escherichia* and *Enterococcus* with age (Supplementary Fig. 4A-B).

## Discussion

In this prospective cohort study of rural Bangladeshi infants monitored with intensive home visits to assess medication use, cumulative exposure to oral antibiotics was high. In this setting, children receiving antibiotics within the previous 7 days had intestinal microbiomes enriched for *Enterococcus* species. However, these changes were not seen in children who had received antibiotics earlier than 7 days prior to collection of the stool sample. Among participants overall, antimicrobial resistance gene carriage fell between ages 8 and 20 months and was correlated with the composition of the microbiome.

Our findings align with previous prospective studies in adult volunteers in high-income settings that showed that oral antibiotics transiently impact gut bacterial load and richness, with duration influenced by antibiotic-class (most transient for the quinolone and cephalosporin, but with recovery within a month for azithromycin)[21]. In infants and children receiving oral antibiotics, several studies have shown acute reductions in alpha diversity and/or specific taxonomic changes including reduced *Bifidobacterium* abundance, though the duration of these changes was not assessed with medium term follow-up (e.g., over weeks)[22,23].

We reasoned that there could be two inter-related explanations for AMR gene abundance: composition and antibiotic use, which may both also be influenced by age. For this reason, we examined relationships between antimicrobial gene abundance and composition at two time points at which we would expect substantial differences in the development of and exposures influencing the microbiome.

Our results indicate that overall, carriage of antimicrobial resistance genes in this cohort of children decreases with age, perhaps relating to maternal transfer of resistant flora that is gradually replaced over time. We identified positive correlations between abundance of both *Enterococcus* and *Escherichia* and the abundance of antimicrobial genes and midline and endline timepoints. *Enterococcus* species exhibit intrinsic reduced susceptibility to some antibiotics and can acquire resistance to many other commonly used antibiotics in this setting[24].

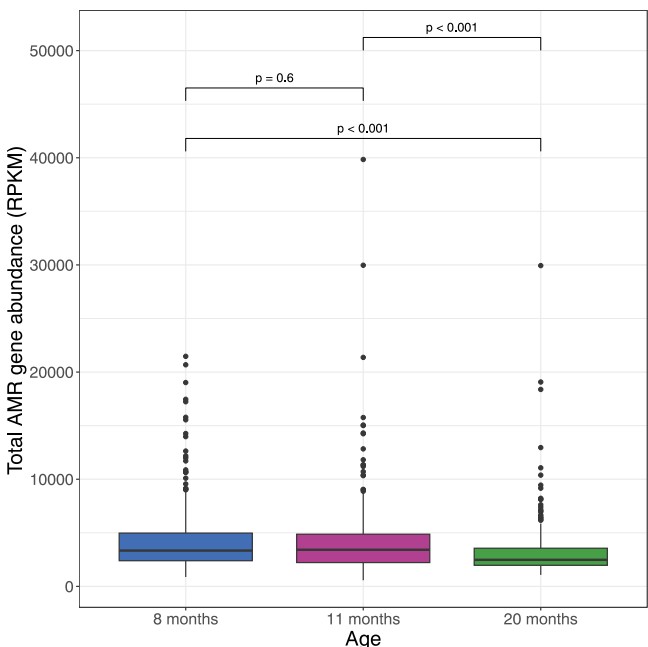

**Fig. 3 | Total antimicrobial resistance gene abundance by age.** Box plot illustrating total antimicrobial resistance gene abundance by age showing statistically significant reductions in total AMR gene abundance between 8 and 20 months of age, and between 11 and 20 months of age ($P < 0.001$ for both comparisons), and not between 8 and 11 months ($P = 0.6$). (Shotgun metagenomic data). $n = 316$ at 8 months, 315 at 11 months and 310 at 20 months. AMR gene abundance was measured in reads per kilobase per million sample reads (RPKM). Group differences were calculated using pair-wise ANOVA. Center lines denote median value, with boxes showing 25–75th percentiles. Vertical whisker lines encompass ×1.5 inter-quartile range from above the upper and below the lower quartiles). Source data are provided in the Source Data file.

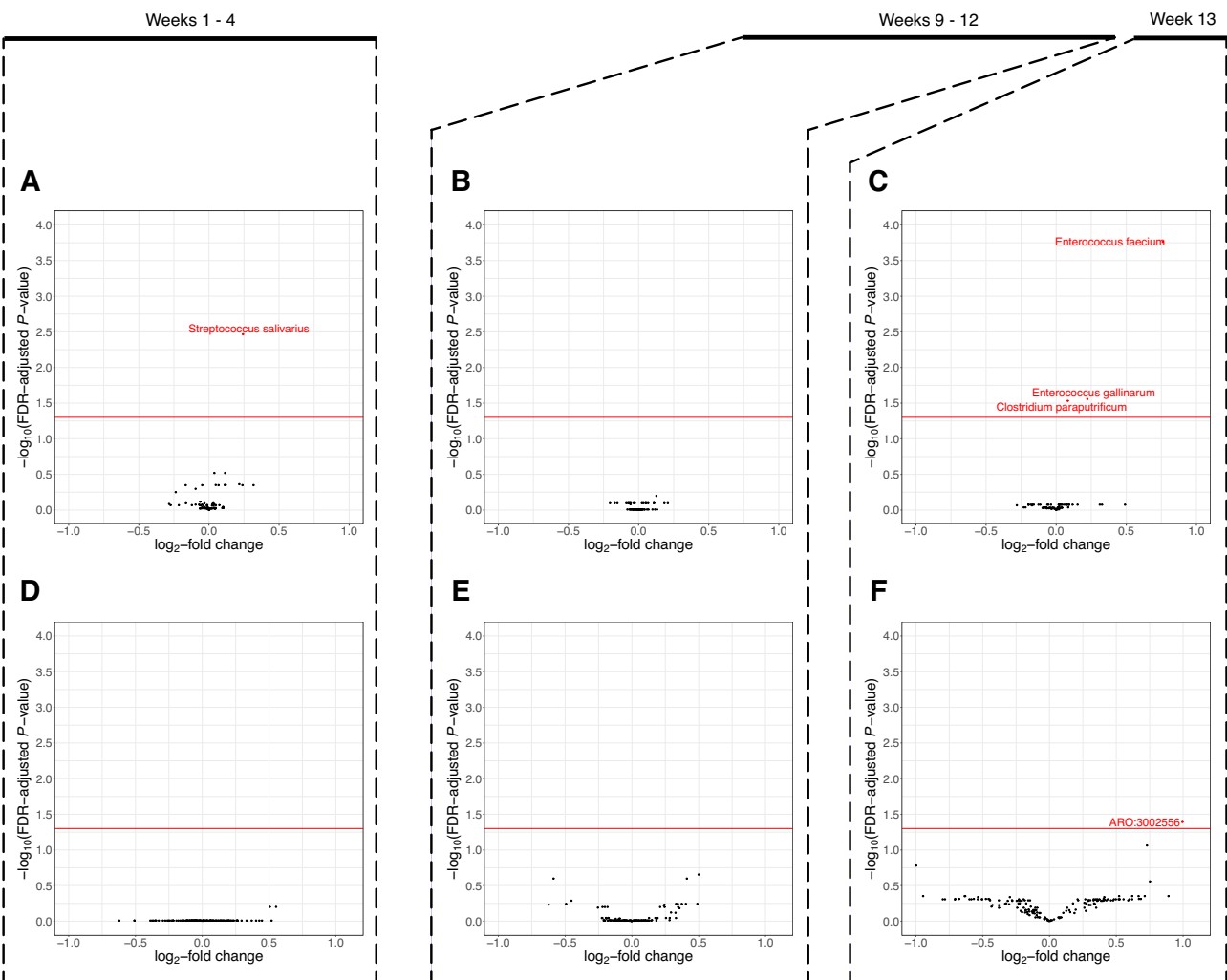

**Fig. 4 | Microbiome and antimicrobial resistance gene differential abundance using shotgun metagenomic sequencing data. A–C** Volcano plots illustrating differential abundance of species in relation to antibiotic use in Weeks 1–4 (38 with antibiotic use from a total of 259 included samples) (**A**), Weeks 9–12 (79/308) (**B**), and Week 13 (18/294) (**C**) (Shotgun metagenomic data). **D–F** Volcano plots illustrating differential abundance of AMR genes in relation to antibiotic use in Weeks 1–4 (38/259) (**D**), Weeks 9-12 (79/308) (**E**), and Week 13 (18/294) (**F**) (Shotgun metagenomic data). Each figure shows $\log_2$-fold change on the *x*-axis and the $-\log_{10}$(FDR-adjusted *P*-value) on the *y*-axis. The horizontal red line marks the level equivalent to an adjusted *P*-value 0.05 (calculated using the Benjamini–Hochberg method). AMR gene abundance was measured in reads per kilobase per million sample reads (RPKM). Source data are provided in the Source Data file.

For example, *Enterococcus faecium* harbors constitutive antimicrobial resistance genes such as aac(6')-Ii, and *E. coli* contains beta-lactamases encoded by genes such as AmpC[25]. In both cases we observed reductions in gut carriage of these species over the 12 month follow-up period of the study. Collectively, these findings suggest that changes in gut flora over time may be a driver of the decrease in abundance of antimicrobial resistance in children in this population.

Our data indicate that use of oral antibiotics in the community may transiently suppress other flora (i.e., reducing alpha diversity) and enrich for these relatively resistant species along with corresponding constitutively encoded antimicrobial resistance genes. Similar findings have been previously reported for example in small cohorts of European infants[26].

A key question is whether recurrent community dosing of antibiotics may eventually select for resistance. Other studies have identified limited impact of antibiotic exposure in children on carriage of antimicrobial resistance genes, particularly evaluating the role of azithromycin. A sub study of a trial of mass biannual administration of azithromycin (a macrolide) that successfully reduced childhood

mortality in Africa revealed that in the Nigerian cohort, the antibiotic did not influence overall microbial structure but macrolide resistance genes were more abundant in children living in intervention communities (although transient increases in resistance to other classes could not be captured by the study design)[27]. Likewise, trials of administering azithromycin during childbirth in The Gambia revealed an elevated prevalence of neonatal carriage of azithromycin-resistant *Staphylococcus aureus*[28] for a period lasting at least 4 weeks but less than 12 months[29]. Notably, the mechanisms of resistance in these studies included the potential horizontal transfer of plasmids encoding resistance genes even to participants not receiving azithromycin. In a multi-country randomized controlled trial of azithromycin versus placebo given to infants and children with acute watery diarrhea, resistance to azithromycin in *E. coli* isolates from stool and *Streptococcus pneumoniae* from nasopharyngeal samples did not differ between groups 90 and 180 days after receiving the intervention. There were also no differences in azithromycin resistance in the equivalent samples and bacteria in household contacts of the study population[30]. Our study provides data on the effects from other classes of antibiotics

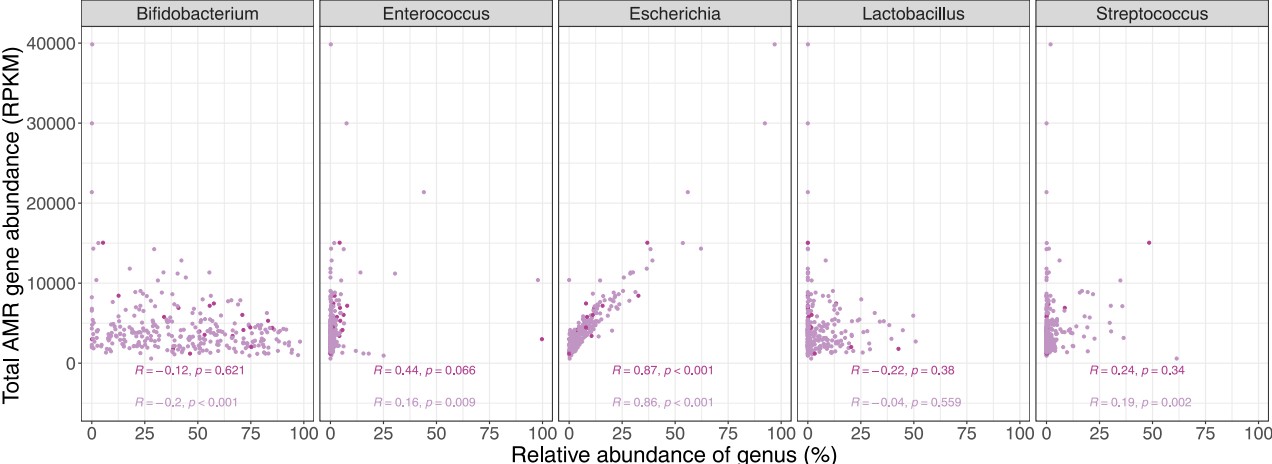

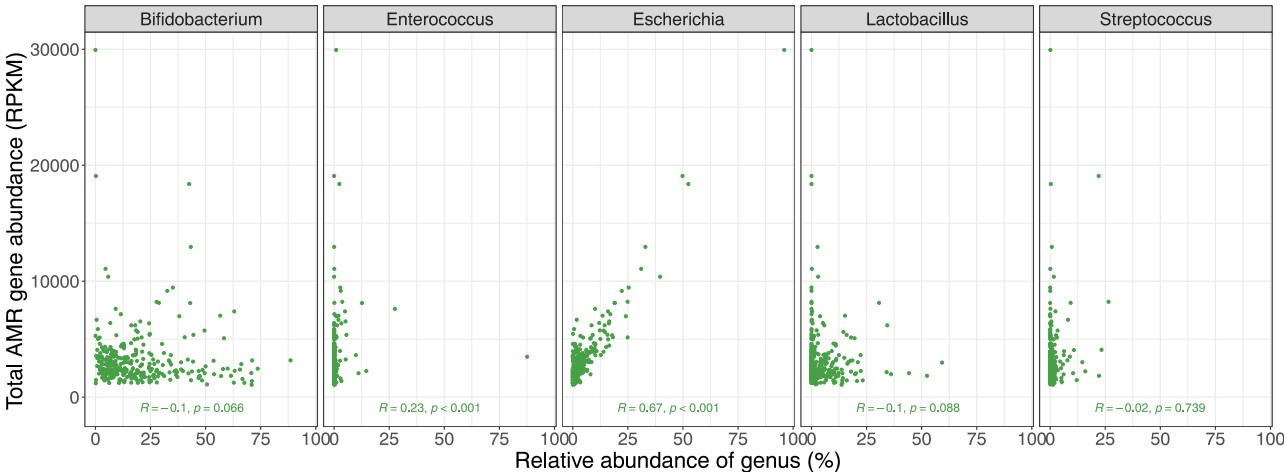

**Fig. 5 | Correlation between antimicrobial resistance gene abundance and microbiome composition. A** Scatterplots illustrating correlation between total antimicrobial resistance gene abundance and relative abundance of five selected genera at midline. Data points, Spearman correlation coefficients and two-sided *P*-values in light purple represent all midline samples with data for both variables and those in dark purple represent samples from participants who received an antibiotic in the preceding 7 days (18 out of 316 available midline samples) (Shotgun metagenomic data). At this timepoint, *Escherichia* showed a highly positive correlation (Spearman *r* = 0.86, *P* < 0.001) with overall AMR gene abundance. *Enterococcus* exhibited a weaker correlation (Spearman *r* = 0.16, *P* = 0.009), although this association appeared to strengthen in children who had received an antibiotic in

the last 7 days: Spearman *r* = 0.44, *P* = 0.066). **B** Scatterplots illustrating correlation between total antimicrobial resistance gene abundance and relative abundance of five selected genera at endline. The green data points, Spearman correlation coefficients and two-sided *P*-values represent all available endline samples (312 available samples) (Shotgun metagenomic data). At this timepoint, total AMR gene abundance and *Escherichia* relative abundance were moderately correlated (Spearman *r* = 0.67, *P* < 0.001). AMR correlation with *Enterococcus* was also observed (Spearman *r* = 0.23, *P* < 0.001) AMR gene abundance was measured in reads per kilobase per million sample reads (RPKM). Source data are provided in the Source Data file.

commonly used in this community setting. In this complex low-income setting, other factors may influence antimicrobial resistance where bacterial infections are common, diagnostic tools limited, care-seeking is variable, and sanitation is lacking leading to high environmental pathogen load and resistance gene sharing[15,16].

Our study is the first to utilize unbiased genomic approaches to define the relationship between community, outpatient use of antibiotics in the local low-income, Asian setting and relate this to the composition and antimicrobial resistance gene profile in young children. Our study confirmed the high usage of antibiotics in children observed in other studies in South Asia[31,32]. We leveraged a clinical trial with intensive home visits for data collection (including photography

of antibiotics to confirm their labeling) and a dedicated stool collection team, and utilized unbiased amplicon and whole-genome approaches to determine the impact of community use of antibiotics on the diversity, composition and antimicrobial resistome profile of the intestinal microbiome in rural Bangladeshi children. This study design is uniquely equipped to provide new information on the impact of community use of antibiotics in this rural, low-income setting, which is a critical global health question.

Our study is likely underpowered for the effects of individual antibiotic classes, and therefore we refrained from conducting class-specific analyses. Antibiotic use – including dose and duration – as reported by the guardian was recorded through home visits, affecting

the enumeration of antibiotic exposure. However, antibiotics were inspected and verified by field workers during these home visits. Samples of the medications were not tested for quality and concentration of antibiotic as this was beyond the scope of the immediate study.

Moreover, our study addressed only the intestinal microbiome; reprofiling and development of antimicrobial resistance may have been more profound and may have been sustained for a longer duration at other sites such as the nasopharynx[29]. Additionally, some children may have already received antibiotics prior to enrollment in the BRISC trial, which may have influenced baseline data. It is unlikely that our data indicate reverse causality, i.e. the changes in gut flora are the reason for antibiotic use, given that most children received antibiotics for fever and respiratory (runny nose, cough) symptoms, with only 21% reporting antecedent diarrheal symptoms. Two large studies that have evaluated the prevalence of diarrhea-causing pathogens in infections in children in LMICs – including Bangladesh – report a range of bacteria, viruses, and parasites responsible[33,34]. Importantly, *Enterococcus* was not highlighted as a bacterial cause of diarrhea in these findings, and it generally does not cause diarrheal infection[35].

In this real-world study conducted in a high antibiotic-usage context in south Asia, community use of antibiotics by rural Bangladeshi infants did not cause sustained reprofiling of the gut microbiome. Our findings do not directly implicate suboptimal clinical antibiotic stewardship and extensive community use of oral antibiotics in the development of antimicrobial resistance in this setting. Under the One Health approach, it is crucial to examine the impact of other sources of excessive antibiotic use, such as their use in the care of livestock, poultry, and fish, as well as hospital settings[16,36]. Evaluating the collective effects of diverse sources of antibiotic exposure is essential for a more thorough understanding of antimicrobial resistance dynamics in this region.

## Methods

We undertook a sub study within the BRISC (Benefits and Risks of Iron InterventionS in Children) (trial registration: ACTRN12617000660381, WHO UTN U111-1196-1125) trial in rural Bangladesh[17]. The trial protocol, including stool collection and microbiome analyses, was approved by ethics committees at the International Centre for Diarrhoeal Disease Research, Bangladesh (icddr,b) and Melbourne Health, Melbourne, Australia; and overseen by an independent Data Safety and Monitoring Board. Parents or guardians of participants provided written consent prior to enrollment and travel costs for study visits were reimbursed. The full protocol is available with the main BRISC outcome publication[17].

BRISC recruited 3300 children aged eight months living in Rupganj Upazila, Bangladesh (a rural area about 50 km from Dhaka) and randomized them 1:1:1 to three months of either daily iron syrup (and placebo multiple micronutrient powders, MNPs); iron, zinc, ascorbic acid, vitamin A and folate containing MNPs (+ placebo iron syrup) or placebo (placebo syrup and placebo MNP). To align with overall trial timelines, we sought to recruit the final 1093 children of the BRISC trial to the microbiome sub study for 16S rRNA amplicon analysis of stool. Samples for shotgun metagenomic sequencing were chosen randomly from the sub study cohort, with priority given to samples from participants who had a baseline and midline (+/- endline) sample available.

### Statistics and reproducibility

As this was a retrospective, cross-sectional analysis of the impact of antibiotic use on the microbiome, sample size was not statistically predetermined. However, the relatively large sample size for 16S rRNA amplicon analysis, combined with the high sequencing depth from shotgun metagenomics, makes this one of the largest microbiome studies undertaken in the field.

Investigators were blinded to group allocation throughout both data collection and analysis. Samples were also sequenced in random order (i.e., PCR plates and sequencing runs included samples from different timepoints) to minimize any batch effects. Samples that yielded <500 reads (16S rRNA amplicon sequencing) and/or that failed the sequencing process were excluded from the analysis. Antibiotic use and reported infections were not exclusion criteria.

### Stool collection and transport

Guardians of all participants of the sub study were educated on stool collection, provided nappies and specimen containers, and asked to collect the stool passed within 3 hours of the visit. If stools were passed after the visit, dedicated staff were sent to retrieve the samples. Samples were stored on ice and transported to the field laboratory within a further 3 hours, where samples were aliquoted and DNA/RNA Shield (Zymo Research) added prior to freezing to −20 °C. Specimens were then transported on dry ice to a −80 °C freezer at the International Centre for Diarrhoeal Disease Research, Bangladesh (icddr,b) facility in Dhaka, Bangladesh, and subsequently shipped on dry ice to The Walter and Eliza Hall Institute of Medical Research (WEHI), Australia.

### Participant data relating to symptoms

Symptoms relating to infection were reported by the mother/guardian as part of home visits by field study staff during intervention period (weekly, recall in the last 7 days) and follow-up period (monthly, recall in the last month). If the participant had no data collected at a particular week, it was assumed that the participant did not have any symptoms.

### Participant data relating to antibiotic use

Antibiotic medication use was also reported by the mother/guardian in the same home visits, and the antibiotic was visually verified by the study worker. We used data from the weekly visits to classify timing of antibiotic use in relation to the midline sampling time point, establishing three definitions: antibiotic use in the week preceding stool sampling, compared to no antibiotic use in that week; antibiotic use in any of Weeks 9-12, compared to no antibiotic use in Weeks 9-12, with any participant who also recorded Week 13 antibiotic use removed from the analysis; and antibiotic use in any of Weeks 1–4 only, compared to no antibiotic use in the intervention period, with any participant who recorded antibiotic use in Weeks 1–4 and in a subsequent week or weeks removed from the analysis. If the participant had no data collected at a particular week, it was assumed that the participant did not take antibiotic medication.

### DNA extraction for microbiome analysis

Aliquots of samples were added to bead tubes (PowerBead Pro) along with lysis buffer and were homogenized on a TissueLyser LT (Qiagen, Venlo, Netherlands) for 10 minutes at maximum speed. DNA was then extracted with the DNeasy PowerSoil Pro Kit (Qiagen, Venlo, Netherlands).

### 16S rRNA amplicon sequencing

An initial PCR reaction used universal primers (Integrated DNA Technologies) targeting the V4 hypervariable region of the 16S rRNA gene to amplify this region (Supplementary Material). A second PCR reaction joined the common overhang sequences introduced in the previous PCR reaction to dual-index barcode sequences (forward and reverse) for sample identification after sequencing. This method was developed by WEHI genomics core facility and has been used successfully for stool microbiome 16S rRNA amplicon sequencing using these PCR primers for the V4 region[37]. Each PCR plate contained at least one positive control (a healthy donor stool that underwent DNA extraction in parallel with sub study samples), one negative DNA extraction control and several PCR negative controls (blanks). Libraries including negative and positive controls were sequenced using the MiSeq instrument (Illumina, San Diego, USA) using a 300 bp paired-end protocol with 600 cycles, at the WEHI genomics core facility.

## Shotgun metagenomic sequencing

Shotgun metagenomic sequencing was performed on a subset of samples (as well as controls) using the QIAseq FX DNA Library Kit (Qiagen, Venlo, Netherlands), with input DNA and reagents used at 0.5x the specified volumes. First, input DNA was enzymatically fragmented for eight minutes to achieve a fragment size of 450 base pairs. This was followed by adapter ligation and 8 cycles of PCR amplification of library DNA. Library concentration was quantified by Qubit™ dsDNA Assay kit (Thermo Fisher Cat#Q32851) and library size was determined using D1000 ScreenTape (Agilent Cat# 5067-5582) and visualized in TapeStation 4200 (Agilent Cat# G2991BA). Equimolar amounts of libraries were pooled and sequenced using NovaSeq (Illumina, San Diego, USA).

## Bioinformatic and statistical analysis

16S rRNA amplicon FASTQ files were processed using the *DADA2* package in RStudio[38]. Data generated from this process were then imported into *phyloseq*[39]. Samples with <500 reads and amplicon sequence variants (ASVs) with zero reads were removed, and ASVs were agglomerated at the genus level prior to analysis. Alpha diversity was performed using the *microbiomeSeq* package in RStudio using Shannon and inverse Simpson indices[40]. Group differences were calculated using pair-wise ANOVA for each diversity measure. Differential abundance at the genus level was evaluated from the 16S rRNA amplicon data using *ANCOMBC* with default settings and adjusted P-value using the Benjamini-Hochberg method to control for the false discovery rate at a level of 5%[41].

Shotgun metagenomic samples were processed using the bioBakery workflows tool, consisting of the default quality control steps from *KneadData* and taxonomic profiling using *MetaPhlAn*[42]. Taxonomic differential abundance in relation to participant variables was performed at the species level using *MaAsLin2* with a minimum prevalence cut-off of 10% (i.e. a species had to be present in at least 10% of samples to be included in the analysis)[43]. Relative abundance data underwent centered log ratio normalization/transformation in *MaAsLin2*, with data for differentially abundant species including coefficient (approximating $\log_2$-fold change), standard deviation, unadjusted and adjusted P-values (using Benjamini-Hochberg method with a significance cut-off of 0.05)[44].

Separately, the *ShortBRED* pipeline was used first to develop markers from version 3.2.8 (2023) of the Comprehensive Antibiotic Resistance Database (CARD)[19] and then to determine the relative abundance of these markers in the *KneadData* outputs[45]. The CARD database defines the Antibiotic Resistance Ontology (ARO) and markers were developed for the AROs relating to determinants of antibiotic resistance. These results were then used in a separate *MaAsLin2* with the same settings as above to evaluate differential abundance at the ARO level in relation to participant variables. We reported reads per kilobase of reference sequence per million sample reads (RPKM), which adjusts for both sample sequencing depth and marker/gene length, and thus normalizes the number of sequence copies to a standard, enabling comparison between conditions. RPKM is an established approach for presenting microbiome composition and AMR data[46–48].

The relationship between genus relative abundance and total ARO reads overall and by antibiotic use was explored using Spearman's correlation analysis. Spearman's correlation coefficients and unadjusted P-values were calculated using *ggpubr*[49].

Heterogeneity in the effect of antibiotic use in the preceding 7 days on differential abundance of AMR genes and microbiome composition between BRISC trial arms was explored by fitting a model including antibiotic use, treatment group (i.e., iron/MNPs and placebo) and the interaction between antibiotic use and treatment group in a separate *MaAsLin2* analysis with the same settings as above applied

and extracting the $\log_2$-fold change, unadjusted and adjusted P-values of the interaction term.

All P-values are two-sided.

## Reporting summary

Further information on research design is available in the Nature Portfolio Reporting Summary linked to this article.

## Data availability

The sequencing data used in this study have been deposited in the NCBI Sequence Read Archive (SRA) database under BioProject accession PRJNA1081952 [https://www.ncbi.nlm.nih.gov/sra]. Source data are provided with this paper.

## Code availability

Code used to process sequencing files (in R for 16S rRNA files and using Python for shotgun metagenomic files), perform statistical analysis and generate figures (in R) is based on the respective packages cited in the manuscript. Custom code was not used. However, code is available from the authors by request.

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

## Acknowledgements

We thank the local field workers for their support, and all participants and their families for their involvement in the study. The authors gratefully acknowledge the WEHI Advanced Genomics Facility for their support and assistance in this work. This work was supported by the Australian National Health and Medical Research Council GNT1103262 (B.A.B.), GNT1159151 (S.P.), GNT1158696 (S.P.), and GNT2009047 (S.P.), and by The Geok Hua Wong Charitable Trust. icddr,b is also grateful to the Governments of Bangladesh, Canada, Sweden, and the UK for providing core/unrestricted support. This work was made possible through Victorian State Government Operational Infrastructure Support and Australian Government NHMRC IRIISS. AB was supported by a Research Training Program Scholarship from the University of Melbourne and a stipend from WEHI.

## Author contributions

A.B. designed the study, led the fieldwork, performed the microbiome experiments, undertook bioinformatic data analysis, interpreted the findings, and wrote the manuscript. S.B. designed the study, analyzed the data, and wrote the manuscript. M.I.H. designed the study, led the

fieldwork, and interpreted the findings. M.B., N.J., S.W., and R.B. assisted in designing and performing the microbiome experiments. A.J. and I.A.A. assisted with bioinformatic data analysis. C.B., G.M.M., R.A., D.C., L.C.H., and A.J. interpreted the findings. M.S.A.B. assisted with fieldwork. B.A.B. and S.E.A. designed the study. S.P. designed the study, interpreted the findings, and wrote the manuscript.

## Competing interests

The authors declare no competing interests.

## Additional information

[1]Population Health and Immunity Division, Walter and Eliza Hall Institute of Medical Research, Parkville, VIC, Australia. [2]Department of Medical Biology, The University of Melbourne, Parkville, VIC, Australia. [3]Centre for Epidemiology and Biostatistics, University of Melbourne School of Population and Global Health, Carlton, VIC, Australia. [4]Department of Infectious Diseases at the Peter Doherty Institute of Infection and Immunity, The University of Melbourne, Melbourne, VIC, Australia. [5]International Centre for Diarrhoeal Disease Research, Bangladesh (icddr, b), Dhaka, Bangladesh. [6]Advanced Technology and Biology Division, Walter and Eliza Hall Institute of Medical Research, Parkville, VIC, Australia. [7]Diagnostic Haematology, The Royal Melbourne Hospital, Parkville, VIC, Australia. [8]Victorian Infectious Diseases Service, Royal Melbourne Hospital, Parkville, VIC, Australia. [9]Faculty of Science, University of Melbourne, Melbourne, VIC, Australia. [10]Clinical Haematology at The Royal Melbourne Hospital and the Peter MacCallum Cancer Centre, Parkville, VIC, Australia.
✉e-mail: Baldi.a@wehi.edu.au; Pasricha.s@wehi.edu.au

