## [Peer Review File · Nature Communications]

REVIEWER COMMENTS

Reviewer #1 (Remarks to the Author):

In this study, Baldi et al. followed the antibiotic usage of 3300 infants (8-months-old over 12 months). 49% of the children were exposed to antibiotics. Using 16sRNA, the authors show that the alpha diversity was decreased in children treated with antibiotics in the 7 days prior testing, with an enrichment in *Enterococcus* and *Escherichia/Shigella*, whereas they did not observe any effect when children were treated earlier than that. Next, they used shotgun metagenomics to measure the abundance of antimicrobial resistance genes overtime.

The authors clearly invested a lot in sample collection and rigorous following of each participant. However, I am not convinced by the general novelty of the project and/or the conclusions from the authors.

General comments

- To me the general question/objective of the project should be more clear.
- There is very few information on how do the nutritional interventions (the used cohort is part of a larger trial on nutritional interventions) affects the microbiome composition of the children? Are there potential confounding factors?
- There is no information about the antibiotics dosage/length of treatment, which would be important to understand the impact.
- I don't think these data offer the right resolution to confidently claim that the amount of AMR genes decreases with age and is not influenced by the individual antibiotic exposure.

Specific comments

- L113 – it is not clear to me whether 3300 or 1093 children participated to this study.
- L165 – I am not sure how the RPKM has been calculated or if it is the best way to analyze the data.
- Figure 5: it seems that in some samples, the relative abundance of *E. coli* is so high that it questions the collection quality.
- The first paragraph is too strong compared to the actual findings of the study

Reviewer #2 (Remarks to the Author):

Baldi and colleagues present intriguing findings supporting the transient effect of self-reported antibiotic use in children enrolled as part of a wider interventional trial. Such studies are relevant for the field of microbiome studies, as antibiotic use is often an exclusion criterion for enrollment and their long-term effect is usually overestimated, as other studies in adults show. Being children more exposed to antibiotics with respect to adults, the findings of this study provide support for the inclusion of infants subjected to treatments weeks before the sampling. Despite the value of the data produced, I have a few comments.

Clarity could be improved greatly by reviewing the wording of some sections. For instance:

a) The introduction flow is pretty clear, but the way the aim / scope of the study is presented is a bit convoluted (Lines 101-109). Is the aim of the study the survey of antimicrobial resistance due to antibiotic use? Is the aim the survey of the effect of antibiotic use on the gut microbiome of children? Is it both? Please, clarify this section.

b) The number of individuals is presented as 3,300, but this is the whole cohort of the interventional trial. It would be better to have the number of infants enrolled for the microbiome study (1093) or the actual number of infants providing samples (923) presented throughout the manuscript.

c) Related to the previous point, Figure 1C-E report data for the whole cohort of 3,300 individuals, but as the study only focuses on the 1093 participants, it would make sense to present data only for the sub-cohort, as it might have different trends wrt the whole cohort

d) The use of both "13 weeks" and "11 months" timepoint is confusing. Please introduce in Line 121 that 13 weeks is the 11-months sampling point, and keep only one between 13-weeks and 11-months throughout the manuscript as this is confusing. The terms "midline" or "end of intervention period" or similars are other options, but please choose one term for this datapoint and homogenize it throughout the text, figures and tables for clarity.

e) The conclusions of the study are not clear as well. What are the main findings and their significance for the field? What are the implications? A rewriting of the Results and Discussion sections to highlight this would help the reader a lot, and probably make their interesting data shine.

Minor points:

- Lines 97-98: what are the rates of resistance for these bacteria in Bangladesh?
- Line 120: please specify that this is an iron supplementation intervention, otherwise it is unclear here what intervention the authors are referring to
- Lines 258-259: are antibiotic use data available for the first timepoint at all?
- Line 310: lacking a period at the end of the sentence

Table 1: in the caption there are typos for the definition of Stunting / Underweight / Wasting

Figure 1A: please add the weekly / monthly visits as breaks in between the main periods, which makes it easier for the reader to understand the study design

Figure 1C-E: these are interesting data, but I would suggest to focus on the cohort that is studied in this manuscript (1093 participants) as the sub-cohort might have different trends wrt the whole cohort

Reviewer #3 (Remarks to the Author):

This is an interesting and relevant exploratory analysis evaluating the complex relationship among antibiotic use, the gut microbiome, and the gut resistome among children living in LMIC settings. The manuscript could be improved by addressing the following comments:

The background focuses almost exclusively on antimicrobial overuse/misuse however this perspective

neglects the very real challenges faced in settings such as Bangladesh where bacterial infections are common, diagnostic tools limited, care-seeking is variable, and sanitation is lacking leading to high environmental pathogen load and resistance gene sharing. The background could be more balanced by including figures on the contribution of poor sanitation and hygiene to AMR etc. See a few articles below:

[https://www.thelancet.com/journals/langlo/article/PIIS2214-109X\(23\)00554-5/fulltext](https://www.thelancet.com/journals/langlo/article/PIIS2214-109X(23)00554-5/fulltext)

[https://www.thelancet.com/action/showPdf?pii=S2542-5196\(23\)00278-4](https://www.thelancet.com/action/showPdf?pii=S2542-5196(23)00278-4)

<https://pubmed.ncbi.nlm.nih.gov/34610785/>

<https://pubmed.ncbi.nlm.nih.gov/28362299/>

Lines 230-232. The ABCD trial offers more AMR-data following azithromycin treatment in both E.coli and Strep pneumoniae 90 and 180-days post-treatment <https://pubmed.ncbi.nlm.nih.gov/34913980/> . See Table 3 and supplementary e-tables 4-7.

Line 262-263 is not true for this setting. There have been two very large diarrhea etiology studies, both of which include sites from Bangladesh, that report bacteria to be a leading cause:

<https://pubmed.ncbi.nlm.nih.gov/27673470/>

<https://pubmed.ncbi.nlm.nih.gov/30287127/>

Many of the associations are found at some ages and not others and this seems to be underdiscussed. Did the authors pre-specify hypotheses related to age-specific associations? If not, why consider the age stratification as it increases risk of type 1 errors.

The authors mention unadjusted and adjusted P-values using Benjamini-Hochberg method with a significance cut-off of 0.05 however it's unclear which p-values are reported (are these adjusted or unadjusted). Was the family-wise alpha set at 0.05?

There is no discussion of the finding that E.coli abundance correlates with AMR genes (line 187-189 in results). What could explain this finding and why do the authors think it's found at 11 months and weaker at 20 months? Given Shigella is part of the Escherichia family, is highly endemic in Bangladesh and is pathogen of major AMR concern, this finding is likely worth discussing in more depth.

Lines 235-236: Could this also not be due to changes in the microbiome (from less diverse to more diverse and from more e.coli to less e.coli which is known to be particularly good at resistance gene carriage)?

REVIEWER COMMENTS

Reviewer #1 (Remarks to the Author):

In this study, Baldi et al. followed the antibiotic usage of 3300 infants (8-months-old over 12 months). 49% of the children were exposed to antibiotics. Using 16sRNA, the authors show that the alpha diversity was decreased in children treated with antibiotics in the 7 days prior testing, with an enrichment in Enterococcus and Escherichia/Shigella, whereas they did not observe any effect when children were treated earlier than that. Next, they used shotgun metagenomics to measure the abundance of antimicrobial resistance genes overtime.

The authors clearly invested a lot in sample collection and rigorous following of each participant. However, I am not convinced by the general novelty of the project and/or the conclusions from the authors.

We thank the reviewer for their feedback and have endeavoured to clarify the value of our project, as well as address the specific comments.

General comments

REVIEWER COMMENT	AUTHOR RESPONSE	MANUSCRIPT REFERENCE
To me the general question/objective of the project should be more clear.	We have added more detail in the Abstract and Introduction to explain that we sought to examine the relationship between antibiotic use and antimicrobial resistance within the gut microbiome. We have added the following text to the Abstract: "We aimed to examine the relationship between oral antibiotic use and composition and antimicrobial resistance in the gut microbiome in 1093 Bangladeshi infants." We have added the following text to the Introduction:	Line 55 Line 112

REVIEWER COMMENT	AUTHOR RESPONSE	MANUSCRIPT REFERENCE
	"We thus sought to determine whether, in a rural South Asian setting, antibiotic exposure through the community reprofiles the gut microbiome and influences carriage of antimicrobial resistance genes in children."	
There is very few information on how do the nutritional interventions (the used cohort is part of a larger trial on nutritional interventions) affects the microbiome composition of the children? Are there potential confounding factors?	We were mindful that our study leverages an iron intervention trial in which two thirds of the participants were exposed to either iron or MNPs and one third to placebo. For this reason, we explored any potential interaction between trial arm (i.e., iron/MNPs versus placebo) and antibiotic exposure (Supplementary Figure 3). As presented, we found no evidence that the effect of antibiotic use on the differential abundance of AMR genes was dependent on iron/MNPs or placebo. In response to this comment we performed an additional analysis of interaction between trial arm (i.e., iron/MNPs versus placebo) and microbiome compositional differential abundance at the genus level, and similarly found no evidence that the effect of antibiotic use on microbiome composition (relative abundance of genera) was dependent on iron/MNPs or placebo.	

REVIEWER COMMENT	AUTHOR RESPONSE	MANUSCRIPT REFERENCE
	 We have added the following text to the Results: "There was also no evidence of an interaction between iron interventions and the effect of antibiotics on microbiome composition (Supplementary Figure 3B)."	Line 196

REVIEWER COMMENT	AUTHOR RESPONSE	MANUSCRIPT REFERENCE
	We have added this figure as Supplementary Figure 3B and updated the corresponding figure legend: "B: Volcano plot illustrating the interaction between BRISC trial arm (iron/MNPs and placebo) and antibiotic use in the preceding 7 days in relation to taxonomic differential abundance at the genus level (Shotgun metagenomic data)."	Supplementary Figure 3B Line 684
There is no information about the antibiotics dosage/length of treatment, which would be important to understand the impact.	Antibiotic use was recorded in weekly and monthly visits and was not a pre-specified primary or secondary outcome of the main larger trial. Aside from drug names, details regarding dosage and duration were not available as these data were obtained from guardian reports during home visits. We have updated the Discussion to reflect this: "Our study is likely underpowered for the effects of individual antibiotic classes, and therefore we refrained from conducting class-specific analyses. Antibiotic use as reported by the guardian was recorded through home visits by field workers, affecting the enumeration of antibiotic exposure. Samples of the medications were not tested for quality and concentration of antibiotic as this was beyond the scope of the immediate study."	Line 294
I don't think these data offer the right resolution to confidently claim that the amount of AMR genes decreases with age and is not influenced by the individual antibiotic exposure.	We have deleted the sentence in the Discussion regarding the role of antibiotic exposure on this change over a 12 month period.	Discussion section

Specific comments

REVIEWER COMMENT	AUTHOR RESPONSE	MANUSCRIPT REFERENCE
L113 – it is not clear to me whether 3300 or 1093 children participated to this study.	We have edited the manuscript to clarify that 1093 children (of the total trial number of 3300) participated in this microbiome sub study.	Lines 57, 124
L165 – I am not sure how the RPKM has been calculated or if it is the best way to analyze the data.	We used RPKM (reads per kilobase of reference sequence per million sample reads) based on the default normalised units calculated by ShortBRED (bioBakery tools); RPKM adjusts for both sample sequencing depth and marker/ gene length, and thus normalizes the number of sequence copies to a standard, enabling comparison between conditions. RPKM is an established approach for presenting microbiome composition and AMR data. Many AMR studies have utilised RPKM values to estimate and compare gene abundance between samples, including the original ShortBRED paper, which used RPKM values produced by ShortBRED to analyse AMR profiles of stool samples obtained from cohorts from four different geographical regions,⁴³ and studies which defined ‘personal microbiome profiles’.¹ (Reference not in manuscript) ShortBRED and RPKM values have also been used in a comparative study of AMR gene profiles in infant gut microbiomes pre- and post-antibiotic administration.² (Reference not in manuscript) We have added the following text in the Methods section: "We reported reads per kilobase of reference sequence per million sample reads (RPKM), which adjusts for both sample sequencing depth and marker/ gene length, and thus normalizes the number of sequence copies to a standard, enabling comparison between conditions. RPKM is	Line 425

REVIEWER COMMENT	AUTHOR RESPONSE	MANUSCRIPT REFERENCE
	an established approach for presenting microbiome composition and AMR data. ^{46-48"}	
Figure 5: it seems that in some samples, the relative abundance of E. coli is so high that it questions the collection quality.	Thank you for drawing our attention to this; we have carefully rechecked the provenance of the high burden samples. There were indeed 3 samples (two at midline, and one at endline) that demonstrated Escherichia relative abundance >90%. The two midline samples had Escherichia relative abundance of 96.92% and 92.31%, with the next highest being 62.2%. The endline sample had Escherichia relative abundance of 96.0%, with the next highest being 52.4%. We have evaluated this further to ensure the quality of our collection. Each of these three samples were collected from different individuals and on different dates, and were processed on different sequencing runs. The negative control samples (water, negative extraction controls) included on each of these sequencing runs did not exhibit contamination with Escherichia. Thus, we are confident there was no systemic factor that may have introduced contamination or bias towards Escherichia. As a sensitivity analysis, we explored removing the data from these three samples with high levels Escherichia to assess the influence of these observations on the overall analysis result. This is presented below: Midline:	

REVIEWER COMMENT

AUTHOR RESPONSE

**MANUSCRIPT
REFERENCE**

Endline:

We note that the result remains statistically significant and has the same effect size as the analysis presented in the manuscript. For this reason, we have chosen to retain the original panel containing the complete data.

REVIEWER COMMENT	AUTHOR RESPONSE	MANUSCRIPT REFERENCE
	We have added the following text in the Results: "A sensitivity analysis following removal of two midline samples and one endline sample with Escherichia relative abundance >90% did not change our results."	Line 210
The first paragraph is too strong compared to the actual findings of the study	We have modified the first paragraph of the Discussion as follows, which we believe accurately summarises the findings of the study: "In this prospective cohort study of rural Bangladeshi infants monitored with intensive home visits to assess medication use, cumulative exposure to oral antibiotics was high. In this setting, children receiving antibiotics within the previous 7 days had intestinal microbiomes enriched for Enterococcus species and exhibited an increased abundance of an associated antimicrobial resistance gene. However, these changes were not seen in children who had received antibiotics earlier than 7 days prior to collection of the stool sample. Among participants overall, antimicrobial resistance gene carriage fell between ages of 8 and 20 months and was correlated with the composition of the microbiome."	Line 215

Reviewer #2 (Remarks to the Author):

Baldi and colleagues present intriguing findings supporting the transient effect of self-reported antibiotic use in children enrolled as part of a wider interventional trial. Such studies are relevant for the field of microbiome studies, as antibiotic use is often an exclusion criterion for enrollment and their long-term effect is usually overestimated, as other studies in adults show. Being children more exposed to antibiotics with respect to adults, the findings of this study provide support for the inclusion of infants subjected to treatments weeks before the sampling. Despite the value of the data produced, I have a few comments.

We thank the reviewer for their feedback and have endeavoured to clarify as well as address the specific comments.

REVIEWER COMMENT	AUTHOR RESPONSE	MANUSCRIPT REFERENCE
Clarity could be improved greatly by reviewing the wording of some sections. For instance: a) The introduction flow is pretty clear, but the way the aim / scope of the study is presented is a bit convoluted (Lines 101-109). Is the aim of the study the survey of antimicrobial resistance due to antibiotic use? Is the aim the survey of the effect of antibiotic use on the gut microbiome of children? Is it both? Please, clarify this section.	We have added the following text to the Introduction to clarify the aims of this study: "We thus sought to determine whether, in a rural South Asian setting, antibiotic exposure through the community reprofiles the gut microbiome and influences carriage of antimicrobial resistance genes in children."	Line 112
b) The number of individuals is presented as 3,300, but this is the whole cohort of the interventional trial. It would be better to have the number of infants enrolled for the microbiome study (1093) or the actual number of infants providing samples (923) presented throughout the manuscript.	We thank the reviewer for this proposal and have omitted presenting the 3300 children unless referring to the design of the parent trial.	Lines 57, 124

REVIEWER COMMENT	AUTHOR RESPONSE	MANUSCRIPT REFERENCE
c) Related to the previous point, Figure 1C-E report data for the whole cohort of 3,300 individuals, but as the study only focuses on the 1093 participants, it would make sense to present data only for the sub-cohort, as it might have different trends wrt the whole cohort	Thank you – we have updated Figure 1C-E and Table 1 to include data from only to the 1093 children included in the microbiome substudy. Related manuscript text was updated accordingly.	Figure 1C-E
d) The use of both “13 weeks” and “11 months” timepoint is confusing. Please introduce in Line 121 that 13 weeks is the 11-months sampling point, and keep only one between 13-weeks and 11-months throughout the manuscript as this is confusing. The terms "midline" or "end of intervention period" or similars are other options, but please choose one term for this datapoint and homogenize it throughout the text, figures and tables for clarity.	We agree with the reviewer and for clarity have changed the term '13 weeks' to 'midline' or '11 months' except when defining the duration of the intervention period.	Figure 1A Line 358
e) The conclusions of the study are not clear as well. What are the main findings and their significance for the field? What are the implications? A rewriting of the Results and Discussion sections to highlight this would help the reader a lot, and probably make their interesting data shine.	We have updated our Discussion in line with the recommendation of the reviewer, and whilst being mindful of Reviewer 1 who cautioned us over overstating the strength of our results. We hope our new draft has struck an appropriate balance.	Discussion section

REVIEWER COMMENT	AUTHOR RESPONSE	MANUSCRIPT REFERENCE
Minor points: - Lines 97-98: what are the rates of resistance for these bacteria in Bangladesh?	We thank the reviewer for this question and have used this as an opportunity to highlight the literature indicating high rates of AMR in Bangladesh among these key bacteria. We have added the following text: "In Bangladesh, studies evaluating clinical isolates have identified pathogenic bacteria such as Salmonella typhi, Escherichia coli and Staphylococcus aureus resistant to commonly used, cost-effective antibiotics, with rates of resistance among the highest in the world.^{12,13} For example, a systematic analysis of the global burden of antimicrobial resistance indicated that in Bangladesh, E. coli resistant to third-generation cephalosporins and/or fluoroquinolones represented 60-70% and ≥80% of isolates respectively, while the prevalence of methicillin resistance in S. aureus is 40-50%.¹⁴"	Line 96
- Line 120: please specify that this is an iron supplementation intervention, otherwise it is unclear here what intervention the authors are referring to	Thank you – we have amended this paragraph for clarification. We have amended the text as follows: " Participants were visited weekly during the 13-week intervention period (iron, multiple micronutrient powders – MNPs – and placebo) and then monthly for 9 months."	Line 132
- Lines 258-259: are antibiotic use data available for the first timepoint at all?	Home visits and thus antibiotic data collection began at the completion of the first week on trial (i.e., after the start of the randomized intervention), consisting of guardian-reported use for the first 7 days post-randomization. We recognise some children may have had distorted microbiomes at the baseline of the trial.	

REVIEWER COMMENT	AUTHOR RESPONSE	MANUSCRIPT REFERENCE
	We have added the following text to the Discussion: "Additionally, some children may have already received antibiotics prior to enrolment in the BRISC trial, which may have influenced baseline data."	Line 302
- Line 310: lacking a period at the end of the sentence	Thank you – this has been corrected.	Line 357
Table 1: in the caption there are typos for the definition of Stunting / Underweight / Wasting	We have corrected typographical errors in the Table 1 footnote.	Table 1
Figure 1A: please add the weekly / monthly visits as breaks in between the main periods, which makes it easier for the reader to understand the study design	We have amended the figure as recommended.	Figure 1A
Figure 1C-E: these are interesting data, but I would suggest to focus on the cohort that is studied in this manuscript (1093 participants) as the sub-cohort might have different trends wrt the whole cohort	We have amended these data to reflect the 1093 children included in the microbiome substudy as suggested.	Figures 1C-E

Reviewer #3 (Remarks to the Author):

This is an interesting and relevant exploratory analysis evaluating the complex relationship among antibiotic use, the gut microbiome, and the gut resistome among children living in LMIC settings. The manuscript could be improved by addressing the following comments:

We thank the reviewer for their input and have endeavoured to clarify as well as address the specific comments.

REVIEWER COMMENT	AUTHOR RESPONSE	MANUSCRIPT REFERENCE
The background focuses almost exclusively on antimicrobial overuse/misuse however this perspective neglects the very real challenges faced in settings such as Bangladesh where bacterial infections are common, diagnostic tools limited, care-seeking is variable, and sanitation is lacking leading to high environmental pathogen load and resistance gene sharing. The background could be more balanced by including figures on the contribution of poor sanitation and hygiene to AMR etc. See a few articles below: https://www.thelancet.com/journals/langlo/article/PIIS2214-109X(23)00554-5/fulltext https://www.thelancet.com/action/showPdf?pii=S2542-5196(23)00278-4 https://pubmed.ncbi.nlm.nih.gov/34610785/ https://pubmed.ncbi.nlm.nih.gov/28362299/	The reviewer makes an excellent point about the complexities of managing infectious diseases in resource-poor settings, as well as other factors contributing to AMR in these environments. We have made use of the suggested references and cited them in the Introduction and Discussion sections. In the Introduction: "More broadly, complex interactions between the environment (including water, sanitation and hygiene conditions, and livestock and wildlife) may influence antimicrobial resistance.^{15, 16} "Therefore, a critical question is whether and how antibiotic use in the community in low-income settings, where exposure to infections and other environmental drivers is intense, directly influences microbial composition and antimicrobial resistance, particularly in children."	Line 105 Line 109

REVIEWER COMMENT	AUTHOR RESPONSE	MANUSCRIPT REFERENCE
	In the Discussion: In this complex low-income setting, other factors may influence antimicrobial resistance where bacterial infections are common, diagnostic tools limited, care-seeking is variable, and sanitation is lacking leading to high environmental pathogen load and resistance gene sharing.^{15,16}	Line 276
Lines 230-232. The ABCD trial offers more AMR-data following azithromycin treatment in both E.coli and Strep pneumoniae 90 and 180-days post-treatment https://pubmed.ncbi.nlm.nih.gov/34913980/ . See Table 3 and supplementary e-tables 4-7.	We thank the reviewer for this relevant citation. We have added the following text in the Discussion: "In a multi-country randomized controlled trial of azithromycin versus placebo given to infants and children with acute watery diarrhea, resistance to azithromycin in E. coli isolates from stool and Streptococcus pneumoniae from nasopharyngeal samples did not differ between groups 90 and 180 days after receiving the intervention. There were also no differences in azithromycin resistance in the equivalent samples and bacteria in household contacts of the study population.³⁰"	Line 270
Line 262-263 is not true for this setting. There have been two very large diarrhea etiology studies, both of which include sites from Bangladesh, that report bacteria to be a leading cause: https://pubmed.ncbi.nlm.nih.gov/27673470/ https://pubmed.ncbi.nlm.nih.gov/30287127/	We again thank the reviewer for drawing our attention to these two important etiology studies. We have removed the following text in the Discussion: "It has been previously reported that most diarrhea in this setting is viral or parasitic."	

REVIEWER COMMENT	AUTHOR RESPONSE	MANUSCRIPT REFERENCE
	We have added the following text in the Discussion: "Two large studies that have evaluated the prevalence of diarrhea-causing pathogens in infections in children in LMICs – including Bangladesh – report a range of bacteria, viruses and parasites responsible.^{33,34} Importantly, Enterococcus was not highlighted as a bacterial cause of diarrhea in these findings, and it generally does not cause diarrheal infection.³⁵"	Line 307
Many of the associations are found at some ages and not others and this seems to be underdiscussed. Did the authors pre-specify hypotheses related to age-specific associations? If not, why consider the age stratification as it increases risk of type 1 errors.	We thank the reviewer for this important question. Yes, we did reason that associations between composition and AMR carriage may change with age, which drove our age-specific analyses. We have added the following paragraphs in the Discussion: "We reasoned that there could be two inter-related explanations for AMR gene abundance: composition and antibiotic use, which may both also be influenced by age. For this reason, we examined relationships between antimicrobial gene abundance and composition at two time points at which we would expect substantial differences in the development of and exposures influencing the microbiome. "Our results indicate that overall, carriage of antimicrobial resistance genes in this cohort of children decreases with age, perhaps relating to maternal transfer of resistant flora that is gradually replaced over time. We identified positive correlations between	Line 234 Line 240

REVIEWER COMMENT	AUTHOR RESPONSE	MANUSCRIPT REFERENCE
	abundance of both Enterococcus and Escherichia and the abundance of antimicrobial genes and midline and endline timepoints. Enterococcus species exhibit intrinsic reduced susceptibility to some antibiotics and can acquire resistance to many other commonly used antibiotics in this setting.²⁴ For example, Enterococcus faecium harbors constitutive antimicrobial resistance genes such as aac(6)-li, and E. coli contains beta-lactamases encoded by genes such as AmpC.²⁵ Collectively, these findings suggest that changes in gut flora over time are a driver of the decrease in abundance of antimicrobial resistance in children in this population."	
The authors mention unadjusted and adjusted P-values using Benjamini-Hochberg method with a significance cut-off of 0.05 however it's unclear which p-values are reported (are these adjusted or unadjusted). Was the family-wise alpha set at 0.05?	Yes, we can confirm that the alpha was set at 0.05 for all FDR-adjusted p-values.	N/A
There is no discussion of the finding that E. coli abundance correlates with AMR genes (line 187-189 in results). What could explain this finding and why do the authors think it's found at 11 months and weaker at 20 months? Given Shigella is part of the Escherichia family, is highly endemic in Bangladesh and is pathogen of major AMR concern, this finding is likely worth discussing in more depth.	We thank the reviewer for identifying this omission and have added the following text to the Discussion: "We reasoned that there could be two inter-related explanations for AMR gene abundance: composition and antibiotic use, which may both also be influenced by age. For this reason, we examined relationships between antimicrobial gene abundance and composition at two time points at which we would expect substantial	Line 234

REVIEWER COMMENT	AUTHOR RESPONSE	MANUSCRIPT REFERENCE
	differences in the development of and exposures influencing the microbiome." "Our results indicate that overall, carriage of antimicrobial resistance genes in this cohort of children decreases with age, perhaps relating to maternal transfer of resistant flora that is gradually replaced over time. We identified positive correlations between abundance of both Enterococcus and Escherichia and the abundance of antimicrobial genes and midline and endline timepoints. Enterococcus species exhibit intrinsic reduced susceptibility to some antibiotics and can acquire resistance to many other commonly used antibiotics in this setting.²⁴ For example, Enterococcus faecium harbors constitutive antimicrobial resistance genes such as aac(6')-li, and E. coli contains beta-lactamases encoded by genes such as AmpC.²⁵ Collectively, these findings suggest that changes in gut flora over time are a driver of the decrease in abundance of antimicrobial resistance in children in this population." Shigella is another important member of the Enterobacteriaceae family. We agree that it is a highly abundant pathogen in Bangladesh, and a recent systematic review of 28 studies found high rates of AMR – particularly multi-drug resistance – among Shigella species in Bangladesh.³	Line 240

REVIEWER COMMENT	AUTHOR RESPONSE	MANUSCRIPT REFERENCE
	Shigella was not able to be distinguished from Escherichia at the resolution of 16S rRNA amplicon sequencing of our participant samples, nor was Shigella detected in our shotgun metagenomic analysis.	
Lines 235-236: Could this also not be due to changes in the microbiome (from less diverse to more diverse and from more e.coli to less e.coli which is known to be particularly good at resistance gene carriage)?	Thank you for this important point. We agree that change in microbiome composition may be a key driver of the reduction in overall AMR gene abundance with age. As presented in Figure 3 of the manuscript, AMR abundance reduces with age. Analysing our taxonomic data at the genus level we see a reduction in Escherichia abundance with age (the reduced relative abundance is statistically significant between ages 8-20 and 11-20 months):	

REVIEWER COMMENT

AUTHOR RESPONSE

MANUSCRIPT REFERENCE

We also see a statistically significant reduction in *Enterococcus* abundance (which in Figure 5 was also shown to positively correlate with AMR gene abundance, though less strongly than *Escherichia*).

***Enterococcus*, however, exhibited an overall lower relative abundance than *Escherichia* in the microbiomes of these children.**

REVIEWER COMMENT	AUTHOR RESPONSE	MANUSCRIPT REFERENCE
	 For Bifidobacterium, there is also a marked decrease in abundance across these three time points (but Bifidobacterium demonstrated no significant correlation with AMR abundance as shown in Figure 5). For Streptococcus, there are no statistically significant differences in relative abundance with age.	

REVIEWER COMMENT

AUTHOR RESPONSE

**MANUSCRIPT
REFERENCE**

REVIEWER COMMENT	AUTHOR RESPONSE	MANUSCRIPT REFERENCE
	 We have added the following text to the Results: "We also observed a reduction in overall abundance of Escherichia and Enterococcus with age (Supplementary Figure 4A-B)." We have added the following text to the Discussion: "Collectively, these findings suggest that changes in gut flora over time are a driver of the decrease in abundance of antimicrobial resistance in children in this population."	Line 211 Line 249

Reference cited in this document and not appearing in manuscript:

1. Franzosa, E.A., *et al.* Identifying personal microbiomes using metagenomic codes. *Proceedings of the National Academy of Sciences* **112**, E2930-E2938 (2015).
2. Lebeaux, R.M., *et al.* Impact of antibiotics on off-target infant gut microbiota and resistance genes in cohort studies. *Pediatr Res* **92**, 1757-1766 (2022).
3. Ahmed, S., *et al.* Prevalence of Antibiotic-Resistant Shigella spp. in Bangladesh: A Systematic Review and Meta-Analysis of 44,519 Samples. *Antibiotics (Basel)* **12**(2023).

REVIEWERS' COMMENTS

Reviewer #1 (Remarks to the Author):

The authors have successfully incorporated most of the comments made. The text reads now better and the objectives are more clearly exposed. However, this study is very descriptive and I am not fully convinced that what they measure is sufficient to support the general message that it conveys (i.e. antibiotics do not affect the children microbiota after 1 week and high-antibiotic consumption does not somehow select for AMR).

Minor comments:

- The first result paragraph could be even more clear with the number of children involved in this sub-study
- Stating that antibiotics do not modify the microbiota composition without knowing the dose of duration of treatment is a "dangerous" claim
- 1249 : " changes in the gut flora are a driver of the decrease in abundance of AMR in children in this population" - I don't believe the authors show that it is the change in microbiota that drives the decrease in AMR

Reviewer #3 (Remarks to the Author):

The authors have addressed all of my concerns.

ADDITIONAL REVIEWER COMMENTS

Reviewer #1 (Remarks to the Author):

We again thank the reviewer for their feedback and have addressed comments as below:

REVIEWER COMMENT	AUTHOR RESPONSE	MANUSCRIPT REFERENCE
The authors have successfully incorporated most of the comments made. The text reads now better and the objectives are more clearly exposed. However, this study is very descriptive and I am not fully convinced that what they measure is sufficient to support the general message that it conveys (i.e.	We thank the reviewer for this feedback. We have amended the text to remove text describing the transient impact of the effects of antibiotics on antibiotic profiling etc. In response to reviewer and editorial feedback we have deleted a portion of the Discussion and retained the remaining text as follows:	

REVIEWER COMMENT	AUTHOR RESPONSE	MANUSCRIPT REFERENCE
antibiotics do not affect the children microbiota after 1 week and high-antibiotic consumption does not somehow select for AMR).	'In this setting, children receiving antibiotics within the previous 7 days had intestinal microbiomes enriched for Enterococcus species and exhibited an increased abundance of an associated antimicrobial resistance gene. However, these changes were not seen in children who had received antibiotics earlier than 7 days prior to collection of the stool sample. Among participants overall, antimicrobial resistance gene carriage fell between ages 8 and 20 months and was correlated with the composition of the microbiome. 'Our study is the first to demonstrate the transient nature of microbiome reprofiling in a low-income community context.'	Line 243

REVIEWER COMMENT	AUTHOR RESPONSE	MANUSCRIPT REFERENCE
- The first result paragraph could be even more clear with the number of children involved in this sub-study	Thank you for this feedback. We feel the sentence ' the final 1093 recruited to the trial (between September 2018 and February 2019) were invited to participate in this microbiome sub study' in conjunction with the amended Figure 1 panels provide sufficient clarity regarding the size of the sub study cohort.	Line 151
- Stating that antibiotics do not modify the microbiota composition without knowing the dose of duration of treatment is a "dangerous" claim"	We thank the reviewer for this comment. We have drawn the reader to this limitation of our study in the Discussion: 'Antibiotic use – including dose and duration – as reported by the guardian was recorded through home visits, affecting the enumeration of antibiotic exposure. However, antibiotics were inspected and verified by field workers during these home visits.'	Line 325

REVIEWER COMMENT	AUTHOR RESPONSE	MANUSCRIPT REFERENCE
- I249 : " changes in the gut flora are a driver of the decrease in abundance of AMR in children in this population" - I don't believe the authors show that it is the change in microbiota that drives the decrease in AMR	We thank the reviewer for this suggestion. We do still believe this mechanism plausible. However, we have softened the language in this sentence in the Discussion: 'Collectively, these findings suggest that changes in gut flora over time may be a driver of the decrease in abundance of antimicrobial resistance in children in this population.'	Line 278